

# A knowledge graph algorithm enabled deep recommendation system

Yan Wang[1], Xiao Feng Ma[2] and Miao Zhu[1]

[1] Teaching Affairs Office, Capital Medical University, Beijing, China
[2] Chengfang Technology Co., Ltd, Guangzhou, Guangdong, China

## ABSTRACT

Personalized learning resource recommendations may help resolve the difficulties of online education that include learning mazes and information overload. However, existing personalized learning resource recommendation algorithms have shortcomings such as low accuracy and low efficiency. This study proposes a deep recommendation system algorithm based on a knowledge graph (D-KGR) that includes four data processing units. These units are the recommendation unit (RS unit), the knowledge graph feature representation unit (KGE unit), the cross compression unit (CC unit), and the feature extraction unit (FE unit). This model integrates technologies including the knowledge graph, deep learning, neural network, and data mining. It introduces cross compression in the feature learning process of the knowledge graph and predicts user attributes. Multimodal technology is used to optimize the process of project attribute processing; text type attributes, multivalued type attributes, and other type attributes are processed separately to reconstruct the knowledge graph. A convolutional neural network algorithm is introduced in the reconstruction process to optimize the data feature qualities. Experimental analysis was conducted from two aspects of algorithm efficiency and accuracy, and the particle swarm optimization, neural network, and knowledge graph algorithms were compared. Several tests showed that the deep recommendation system algorithm had obvious advantages when the number of learning resources and users exceeded 1,000. It has the ability to integrate systems such as the particle swarm optimization iterative classification, neural network intelligent simulation, and low resource consumption. It can quickly process massive amounts of information data, reduce algorithm complexity and requires less time and had lower costs. Our algorithm also has better efficiency and accuracy.

# INTRODUCTION

Recommendation systems function according to a user's historical behavior. Big data and other technologies obtain information by deep mining the user's interest preferences, and, in turn, provide the user with personalized product recommendations (*Shepitsen et al., 2008*). Online education platforms such as Udemy and Coursera provide learners with a huge amount of learning resources, removing the constraints of time, space, location, and other conditions (*Zhang, Zhou & Zhang, 2009*). Online learning portals are inherently broad and extensive, and learners can waste time and energy trying to select appropriate learning resources from a vast amount of choices. This may frustrate a user and reduce

Corresponding author
Miao Zhu, zhumiao@ccmu.edu.cn

their motivation and initiative to continue learning. It could easily cause learning disorientation and information overload.

In 1997, *Resnick & Varian (1997)* proposed personalized recommendation technology. With the advent of AI and deep learning, recommendation technology has been successfully applied in the fields of social network recommendation (*Wang et al., 2018*), e-commerce (*Lei et al., 2021*), advertising (*Liu et al., 2021*), Netflix movie recommendation (*Guy et al., 2009*), *etc*. This has brought great economic benefit to the industry, and now recommendation systems have formed an independent discipline. Google released Google Knowledge Graph (KG) technology in 2012 to improve the quality of semantic searches, marking an important milestone in the development of personalized recommendation technology. In the field of education, scholars have explored and studied personalized learning resource recommendation methods and models, and achieved good research results. In *Resnick et al. (1994)*, an adaptive personalized recommendation model was proposed, which used an ontological approach for semantic discovery and a preference and relevance-based approach to rank the relevance of learning objects, learning content, and preferences. The model provided learners with appropriate learning resources. In *Sheng (2020)*, a semantic model and preference-aware service mining method based on user interest point recommendations was proposed to recommend personalized services to learners based on learning behavior and trajectory sequence prediction (*Jia, 2021*). The personalized adaptive learning model constructed by *Wang, Tsai & Lee (2007)* has been used for analyzing learners' navigation access data and learners' behavior patterns, and provides personalized services based on their personal characteristics.

Collaborative filtering recommendations (*Zhu et al., 2017*; *Sweta & Lal, 2017*) provide personalized recommendations based on a user's item rating matrix. However, it has low recommendation accuracy and sparsity problems. Content-based recommendations (*Yuan, Rong & Zhou, 2017*) make recommendations by calculating the similarity between item features and user interest models. However, this model suffers from problems such as system cold start and recommendation accuracy. Rule-based recommendation (*Kong, 2017*) analyzes the relationship between user and item interests based on user browsing history data and formulates recommendation rules, but it also has the problem of system cold start. Currently, knowledge graph-based recommendation systems have received increasing attention from the industry and academia. Knowledge graph-based systems have the advantage of various recommendation techniques, which belong to hybrid recommendation systems (*Sun & Wang, 2012*) and which compensate for the shortcomings of a single recommendation model. During processing, entities and relationships are first represented as knowledge representation vectors (the process is known as knowledge graph representation learning, or KGE), then the match score is computed (recommendation probability) between the target users and candidate items (*Wu, 2007*). TransE, TransD, TransR, TransH and other models are typical representations of this process. Graph structure-based models supervise the representation learning of knowledge features with the topology of graphs and adopt the architecture of the graph neural network (GNN) to learn entities and relationships; typical models include the RippleNet, KGCN, and KGAT models.

Some scholars conduct research through learning pathways. In *Liu & Zhou (2009)*, the influence of situational perception factors on learning path selection was elucidated. *Xia & Ye (2021)* focused on the influence of learners' ability to absorb factors based on recommended paths and designed a personalized learning path generation framework. *Huang & Zhao (2015)* used knowledge graphs to combine learning preferences, learning styles, and other factors to recommend learning paths to learners. In *Jiang & Zhao (2015)*, a multi-agent based personalized recommendation system was designed using a genetic algorithm screening strategy to generate learning paths.

A personalized recommendation system is an effective way to solve the information overload and learning maze of online learning. It can provide personalized recommendations based on users' interests and preferences and further optimize the recommendation effect through interaction with users. However, personalized recommendation systems face many challenges in their applications, such as semantic understanding and contextual modeling. Recommendation systems require a deep understanding of user language expression and contextual information. However, the complexity and ambiguity of natural language make it challenging to accurately understand a user's intentions. At the same time, it is also a challenge to accurately capture and model contextual information for conversations. A sentence may rely on previous conversation history for an accurate interpretation, and traditional recommendation methods are difficult to effectively handle this. Sparsity and cold start issues in existing recommendation systems means that the historical behavior data of users may be relatively sparse, especially when the number of interactions between users and the system is limited. In addition, for new users or new contexts, traditional recommendation methods may encounter cold start issues, resulting in poor recommendation performance.

With the continuous enrichment of network resources and the increasing complexity and diversity of personalized recommendation scenarios, as well as the diverse learning needs of learners, recommendation systems face more and more challenges, mainly including data sparsity, semantic mismatch, homogenization of recommendation results, the need to improve recommendation accuracy, and the lack of interpretability of recommendation results. Reflected in the following aspects: Firstly, how to accurately understand the needs of learners is a key issue. The needs of learners may be specific to specific topics, content, difficulty, *etc*. Therefore, recommendation systems need to accurately capture the needs of learners through various data analysis and learner feedback. Secondly, it is extremely challenging to recommend suitable learning resources based on the needs and interests of learners. The quality, depth, and breadth of learning resources can all have an impact on the learning outcomes of learners, and recommendation systems need to comprehensively consider these factors to make recommendations. Thirdly, in the process of recommending learning resources, how to comprehensively study the correlation between the learning process and learner characteristics, domain knowledge, and learning behavior is also a challenge faced by recommendation systems. Finally, the timeliness of learning resources is also an important issue. The content and knowledge of learning resources are updated quickly, so

recommendation systems need to update and adjust recommendation strategies in a timely manner to ensure the timeliness of recommended learning resources.

The outline of this work is as follows. "Knowledge graph-based recommendation technology" briefly presents the theoretical formulation for recommendation system. "Personalized recommendation algorithm based on deep learning of KG" focuses on a research on deep recommendation system based on knowledge graph algorithm, introduces a convolutional algorithm to decompose the processing process into four stages: feature extraction unit, recommendation unit, cross unit, and knowledge graph unit, in order to improve the quality and efficiency of personalized learning resource recommendations. The numerical examples represent our experimental analysis on algorithm efficiency and accuracy in "Experimental analysis". Finally, some major conclusions are given in "Conclusion".

# KNOWLEDGE GRAPH-BASED RECOMMENDATION TECHNOLOGY

Recommendation technology uses the key information traces of users during Web browsing, mines user preferences, determines their relevance, and uses intelligent recommendation algorithms to select personalized resources to push to users (Fig. 1).

Figure 1 briefly illustrates the business process relationship of the recommendation system, which consists of three main steps: data collection, user interest mining, and personalized result recommendation.

## Traditional recommendation system algorithms

### Collaborative filtering recommendation algorithm

The collaborative filtering recommendation algorithm is a mainstream personalized recommendation algorithm that uses information about the common interests and experiences of groups of people to filter information through cooperative techniques among their agents, viewpoints, and data sources. The system then automatically recommends useful information to users, who filter and judge information according to their preferences.

User-based collaborative filtering algorithm

The user-based collaborative filtering algorithm (UBCF) mainly relies on three links. These include mining the historical interaction behavior data of users and goods, quantitatively evaluating and establishing a quantitative matrix; calculating the similarity of user behavior using Formula (1), and establishing a set:

$$sim(u, v) = \frac{\sum_{s-1}^{n} r_{u,s} r_{v,s}}{\sqrt{\sum_{s-1}^{n} r_{u,s}^2} \sqrt{\sum_{s-1}^{n} r_{v,s}^2}} . \tag{1}$$

Once the data are collected, the algorithm determines the products that exceed the threshold and push the items with ratings above a threshold to the targeted user. This method can perfectly exploit the hidden attributes among users. The recommendation model is shown in Fig. 2.
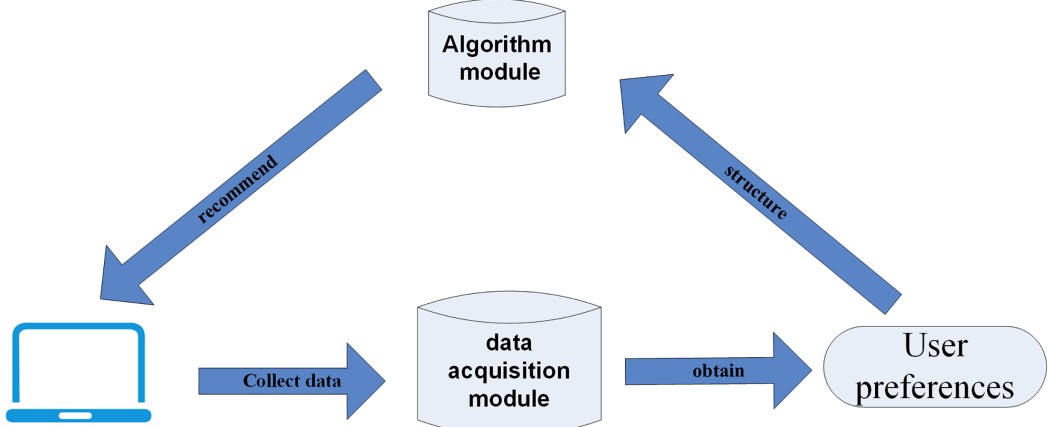

**Figure 1  Flow chart of recommendation system.**

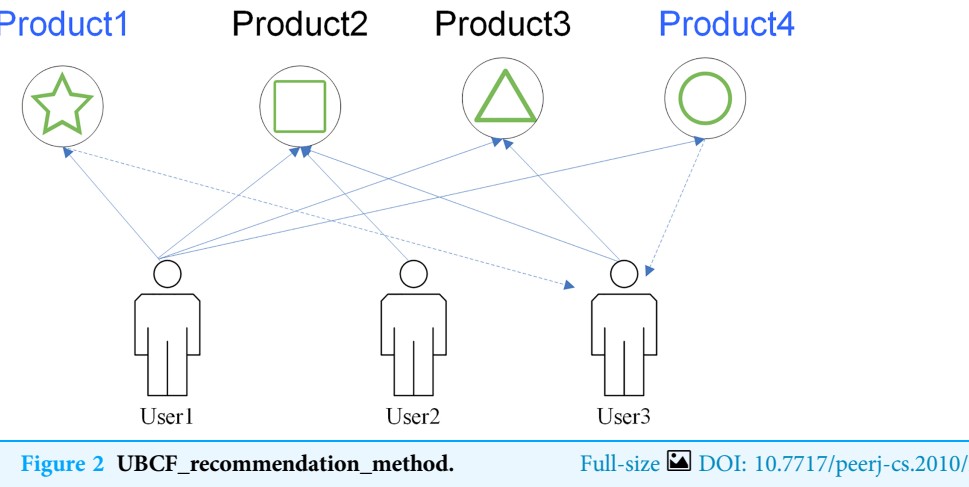

**Figure 2  UBCF_recommendation_method.**

In Fig. 2, all users have browsed the same product, and according to the data association relationship; product 1 through product 4 have similar attributes, so it is clear that users have similar preferences for each other, then the product can be recommended to a specific user.

$$sim(u, v) = \frac{\sum_{s-1}^{n} r_{u,s} r_{v,s}}{\sqrt{\sum_{s-1}^{n} r_{u,s}^2} \sqrt{\sum_{s-1}^{n} r_{v,s}^2}}. \tag{2}$$

Model-based collaborative filtering algorithm

This algorithm is a learning-based algorithm, similar to a neural network algorithm, with the advantages of good robustness and a fast recommendation process. A parametric model is used to establish the association between users and products, and machine learning techniques are used to obtain a predicted list of recommended resources by establishing and modeling the objective function and successively optimizing the objective

function. In general, the algorithm uses a combination of implicit semantics and artificial intelligence for modeling, builds a data matrix of information resources, and trains the data to obtain prediction data. The objective function is:

$$\text{Loss} = \sum_{i=1}^{m} \sum_{j=1}^{n} \left(R_{ij} - P_i^T Q_j\right)^2.\tag{3}$$

In Formula (3), the scoring matrix is $R$: $R = m \times n$; $P$ is the lower dimensional user representation matrix; $Q$ is the item representation matrix.

Collaborative filtering is an excellent recommendation algorithm. However, there are some drawbacks to its practical application. First, it relies too much on the score value; second, the low scalability of the data set sometimes affects the recommendation effect; finally, the lack of historical information is unfavorable to new users and leads to poor recommendation effect. In view of the existing problems, some scholars began to consider introducing other elements into the algorithm and optimizing it to compensate for its shortcomings.

### Content-based recommendation algorithms

Content-based recommendation algorithms (CBR) processes item content to obtain corresponding personalized recommendation resources. The core idea is to perform semantic analysis and modeling based on the attribute features of items, and to recommend items related to the attribute features of entities. Compared with the collaborative filtering-based algorithm, this algorithm has some inherent advantages. Taking the recommended product attributes as input parameters, the algorithm automatically uses the machine learning function to obtain the interconnectedness of products, combines the biased data information of learners individually, and selects similar products with high matching degree to the learners. In this algorithm, the user's evaluation score of the product is no longer the focus, but the matching degree between the learner and the product to be recommended is dynamically calculated around the product features, learners' or users' interests carried out in the product evaluation process. Common algorithms include neural networks, decision trees, and other methods, and their models are shown in Fig. 3.

### Recommendation algorithms incorporating knowledge graphs

Knowledge graphs (KG) are a kind of graph-based data structure with rich semantic relationships. They use information technology such as big data to mine structured, unstructured, and semi-structured data. The knowledge base is then reconstructed in the form of graph through the process of knowledge extraction, fusion, and processing, and personalized recommendation are obtained. It has become one of the key technologies of recommendation systems due to its ability to improve recommendation efficiency. The construction process is shown in Fig. 4.

The knowledge graph consists of nodes and edges, that respectively correspond to "entities" and "relationships" in reality. Usually represented by a triad structure, *i.e.*,

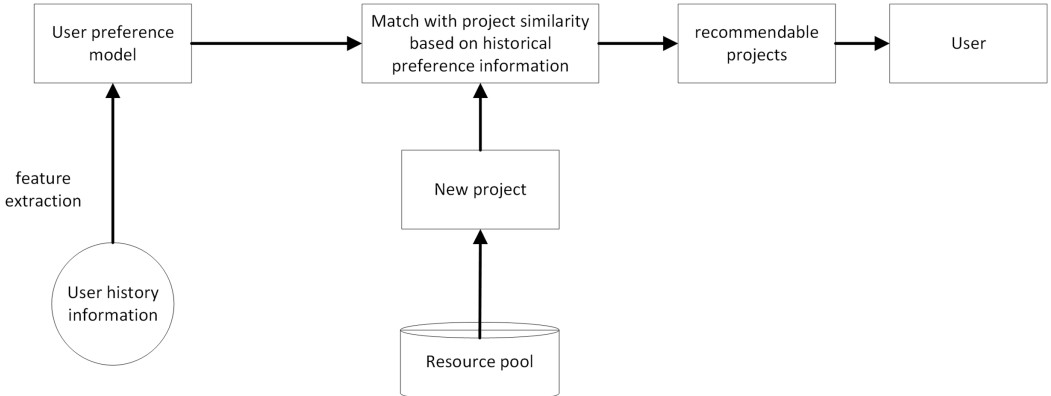

**Figure 3** **Content based recommendation model.**

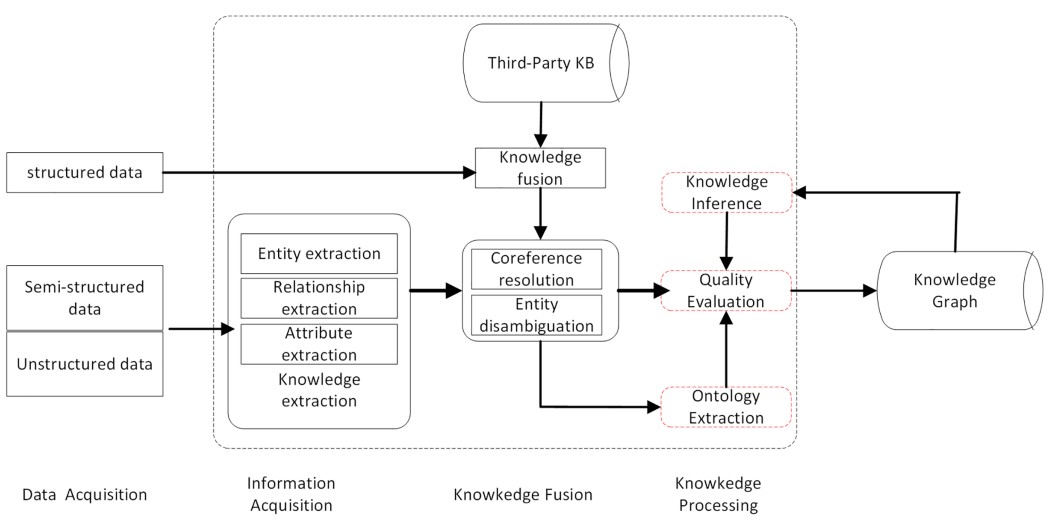

**Figure 4** **KG construction process.**

$G = (E, R, S)$, where E represents entities, R represents relations, and S represents knowledge triads, denoted as (entity, relation, entity), which is $(h, r, t)$. Entities are the node data and are the most basic elements of the KG. Each entity can be represented using a globally determined ID. The knowledge graph contains rich nodes, and a fundamental key aspect of the knowledge graph in the application process is the feature vector or embedding representation of the nodes, where each node is learned to obtain a low-dimensional embedding representation and ensure that this vector contains their location information in the graph as well as the structural information of the neighborhoods in the local graph, as shown in Fig. 5.

Scholars have introduced techniques such as neural networks, convolution, and automatic coding networks based on knowledge graph properties, which are applied in personalized recommendation systems, including the recommendation of learning resources, to improve the recommendation accuracy. Among many recommendation

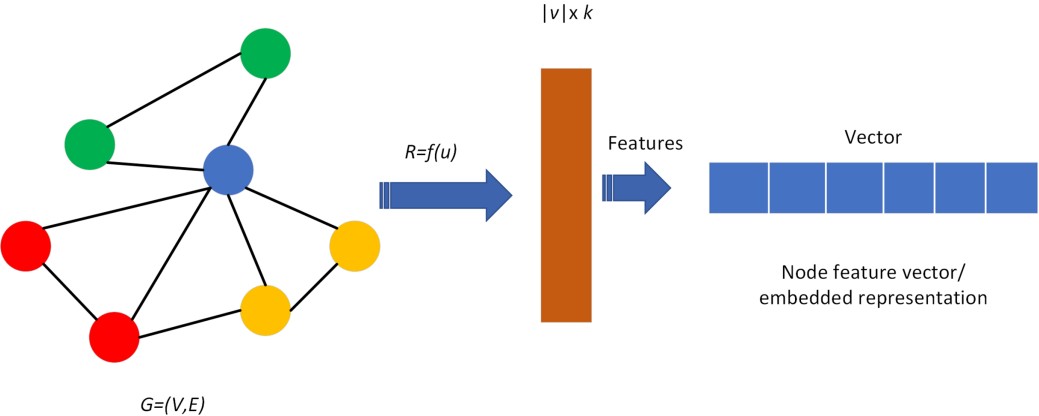

**Figure 5 Node or edge mapping feature vectors in the graph.**

algorithms, the path-based learning method is a superior algorithm, mainly using PathSim (*Sun et al., 2011*; *Ma et al., 2019*) for similarity calculation, which uses the connected path relationship between entities and then uses Eq. (4) to calculate the connected similarity of users and items for recommendation.

$$sim_{m.n} = \frac{2 \times |\{p_{m \to n} : p_{m \to n} \in P\}|}{|\{p_{m \to m} : p_{m \to m} \in P\}| + |\{p_{n \to n} : p_{n \to n} \in P\}|} \tag{4}$$

In Eq. (4), the path between entities is denoted by $p_{x \to y}$.

The recommendation system is fully capable of incorporating the knowledge graph, which allows it to adapt to various application scenarios with highly parsable semantics to improve recommendation accuracy. With its specific network structure, the knowledge graph can be integrated with item and user data to introduce more comprehensive semantic relationships between users and items, and can more accurately mine user interests when the neural network is incorporated.

## PERSONALIZED RECOMMENDATION ALGORITHM BASED ON DEEP LEARNING OF KG

In this section, a personalized learning resource recommendation algorithm model for deep learning is proposed by integrating neural networks and convolutional knowledge graph technology. The model uses deep neural networks to preprocess the original features of users and projects, integrates the vectorization processing of user attributes, conducts embedded learning on the semantic information of knowledge map relations, and obtains the reconstructed knowledge map matrix. After experimental verification, the model was found to have good results in recommendation accuracy.

### Question definition

The recommendation system mainly includes two datasets: user $U$ and project $V$, represented as $u \in U\{u_1, u_2, u_3 \cdots, u_m\}$ and $v \in V\{v_1, v_2, v_3 \cdots, v_n\}$. We first establish a user and project matrix $m \times n$, represented by $Y \in R_{m \times n}$. If there is an association between

users and projects, then $y_{uv} \in Y$ and $y_{uv} = 1$, otherwise $y_{uv} = 0$. Each project v is assumed to have n eigenvalues, and in the triplet $(e_h, r, e_t)$ set of the knowledge graph, the correlation between users and projects is calculated using the formula $\overset{\Lambda}{\underset{uv}{y}} = F(u, v|\theta, Y, G)$.

## Convolutional neural network

Graph convolutional neural networks perform convolution operations on graphs, which have rich semantic correlations in knowledge graphs and are conducive to discovering implicit relationships between items and users. At the same time, the KGE method is used to preprocess KG, mapping entities and relationships into low dimensional representation vectors, reducing time complexity and improving recommendation efficiency. The graph convolutional algorithm is mainly implemented through two methods: frequency domain and spatial domain. The frequency domain method uses Fourier transform to achieve convolution operation of node mapping to frequency domain space. The core of the algorithm is the graph convolution operator, which is calculated by Eq. (5).

$$h_i^{l+1} = \sigma \left( \sum_{j \in N_i} \frac{1}{c_{ij}} h_j^l w_{R_j}^l \right) \tag{5}$$

Graph convolutional network (GCN) modeling is used to obtain higher-order neighborhood information between users and items. A function $\pi_r^u$ is also used to describe the preferences of different users for different relationships, and a formula $g: R^d \times R^d \to R$ (*e.g.*, in the form of an inner product) is used to calculate the score between users and relationships, as defined below:

$$\pi_{r(e^v, e_i)}^u = g \left( u, \sum_{r \in \{R_i\}} r \times \frac{1}{\sqrt{2\pi r}} \exp \left( -\frac{(h_i - \mu)^2}{2\sigma^2} \right) \right) \tag{6}$$

where both u and r are vector representations $r \in R_v$, $u \in R^d$ and $r \in R^d$ the relation represented by $r$, and $d$ is the dimensionality of the vector representation. The shape of the Gaussian distribution can be defined as $h \sim N\left(0, \frac{1}{2\pi}\right)$.

The number of elements in the set $N_{(v)}$ is not fixed in the knowledge graph due to the varying number of neighbors of the nodes. To ensure the fixity and validity of the computational model of the batch process, it is necessary to unify the length of the set of neighbors of the central node, instead of using the set of Kai complete alternative data. The domain of entity $v$ is denoted as $V_{N(v)}^u$, $E_v$ for the single level feeling domain of entity $v$. In order to represent the topology of the wandering sequence of item $v$, the linear combination of the wandering sequence of item v is calculated by Eq. (7).

$$v_{N(v)}^u = \sum_{e \in E_v} \sum_{r \in R_v} \tilde{\pi}_{r,v}^u e, \quad N_{(v)} = E_v \cup R_v. \tag{7}$$

In Eq. (7), $r$ and $e$ must exist in a triad, and if there are $e$ connected to the relation $r$ in the set of attributes, the first entity is chosen as the multiplication term of the formula. In

the set the entities and relations are strictly sorted according to the wandering order, therefore, the order of the entities is related to their occurrence probability, and the higher the occurrence probability, the higher the entity node is in the set $E_v$. $\tilde{\pi}^u_{r,v}$ is the normalized user relationship score, which is calculated by Eq. (8).

$$\tilde{\pi}^u_{r,v} = \frac{\exp(\pi^u_r)}{\sum\limits_{r \in R_v} \exp(\pi^u_r)}. \tag{8}$$

The final step of the KGCNpro layer is to aggregate the entity representation $v$ and its neighbor representations $V^u_{N(v)}$ into a vector. The three types of aggregator representations are shown below. Aggregate the representation of entity $v$ and its wandering sequence representation $V^u_{N(v)}$ into a vector.

$$agg_{sum} = \sigma(W \bullet (v + v^u_{N(v)}) + b)$$
$$agg_{concat} = \sigma(W \bullet concat(v + v^u_{N(v)}) + b) \tag{9}$$
$$agg_{neigbour} = \sigma(W \bullet v^u_{N(v)} + b)$$

To improve computational efficiency, a negative sampling strategy is used in the training process. The full loss function is as follows:

$$L = \sum \left( -\sum_{v:y_{uv}=1} H(y_{uv}, \hat{y}_{uv}) - \sum_{i=1}^{T^u} E_{v_i:p(v_i)} H(y_{uv}, \hat{y}_{uv}) \right) + \lambda ||F||^2_2. \tag{10}$$

In the above equation, $H$ is the cross-entropy loss function, $p$ is the negative sample distribution, and $Tu$ is the number of samples negatively sampled by the user, where $p$ is subject to a uniform distribution.

## Learning algorithm

In the process of implementing the knowledge mapping algorithm, there is a need to consider the topology and balance the relationship between entities and relational nodes of the quantification process. For some characteristic types of data, mining by the knowledge mapping approach is not quite suitable, and it needs to be processed separately by using multimodal ideas. When the knowledge graph approach is used to deal with topological structure data, low semantic parsability occurs, leading to the phenomenon of incomplete information. In this study, we propose a deep recommendation system algorithm based on knowledge graph (D-KGR), which introduces a cross-compression link in the feature learning process of knowledge graph and the feature learning process of item user attributes, optimizes the item attribute processing process by using multimodal technology, processes text-type attributes, multi-value-type attributes and other types of attributes separately, and reconstructs the knowledge graph (Fig. 6) shows the deep network structure of the D-KGR model, which has several data processing units: recommendation unit (RSU), knowledge graph feature representation unit (KGE), cross-compression unit (CCU), and feature extraction unit (FEU).

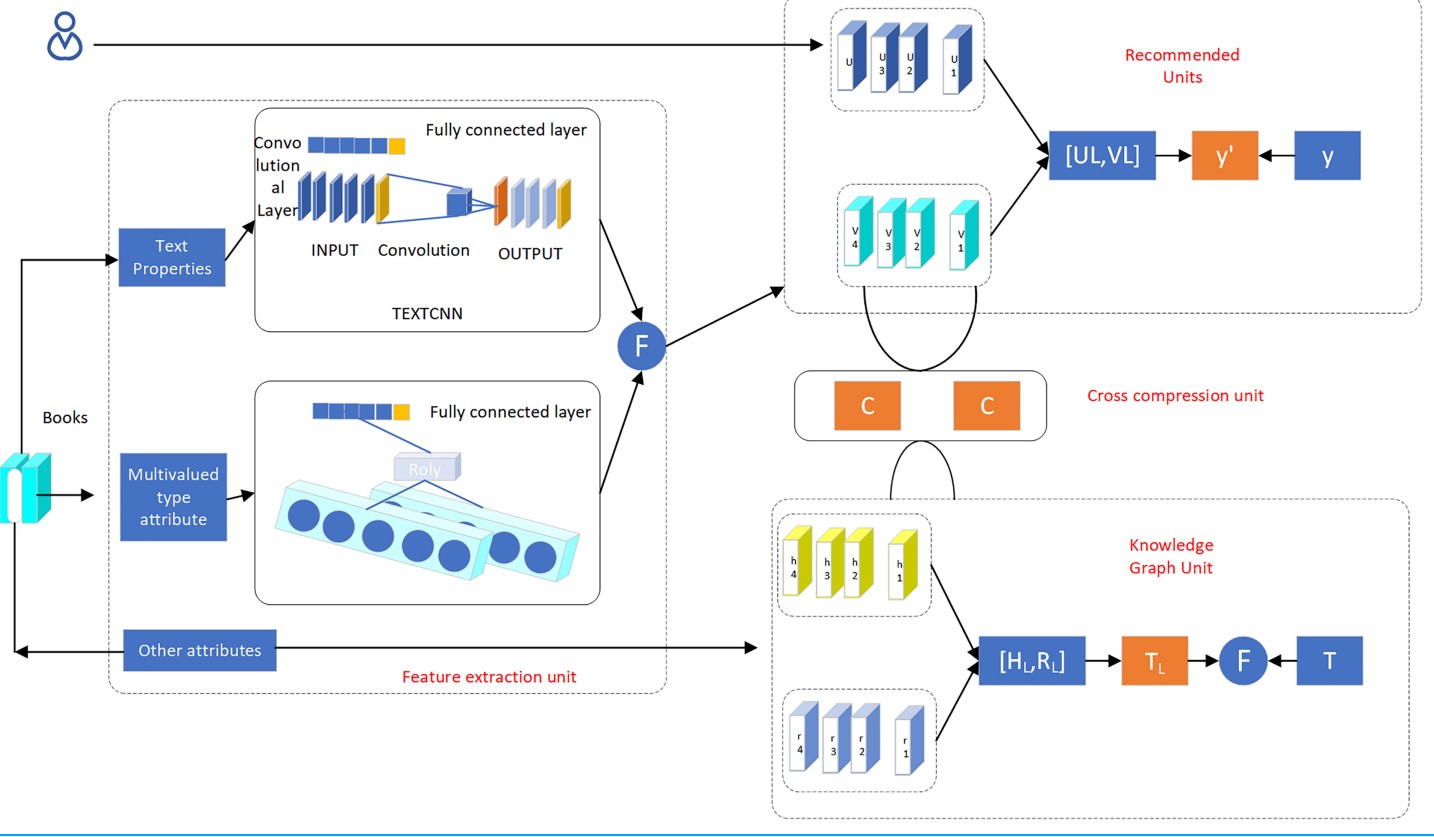

**Figure 6 Deep network structure of D-KGR model.**

## FEU unit

The conversion of attribute relationships between items into the form of corresponding knowledge graph triads is a key step in the process of knowledge graph processing. The conversion process may result in the phenomenon of data attribute value loss and affect the accuracy of the recommendation. In data feature extraction, we optimize the traditional data processing mode of emphasizing macro data structures and ignoring the basic information of micro data, adopt multimodal thinking, classify the basic information of item attribute data, focus on the accuracy of knowledge graph structure information mining, strengthen the semantics of processing entities themselves, and extract the important data attributes with representative meaning for feature extraction. Sets of item attributes may be divided into three types: text, multi-value conforming attributes and other types of attributes. There are differences in the data processing methods.

Convolution is an important signal processing method widely used in fields such as image processing, speech recognition, and natural language processing, making it important for machine learning and artificial intelligence applications. Its main functions include feature extraction, dimensionality reduction, denoising, image enhancement, and indiscriminate compression of data to improve algorithm performance. Feature extraction is part of data processing and convolution can extract features in the signal, such as edges,

textures, *etc.*, through filters. This is very important for image classification and recognition tasks. Convolution is also important in dimension reduction as it can reduce the size of images through pooling operations, thereby reducing the dimensionality of data, which is very useful for processing large-scale image and text data. Denoising and convolution can remove noise from signals through filters, which is very common in the fields of signal processing and image processing and helps to improve the quality of data. Convolutional filtering is helpful for image processing because it can enhance images through filters, which can help improve the visual effect and quality of the image. Convolutional neural networks (CNN) are currently one of the most important models in deep learning to improve algorithm performance, with their basic structure being convolutional layers. Convolution can reduce the size of data through operations such as dimensionality reduction and filtering, thereby achieving data compression. This is very useful for processing large-scale data, achieving data storage and transmission.

Text type data has the characteristic of structural singularity, and in the process of text feature extraction, the convolutional network algorithm is introduced in Markov networks or recurrent neural networks and other methods to effectively reduce the iterative update parameters and achieve the best computation speed with a small amount of computation resources. Assuming that a word is considered as a feature vector of dimension $k$, a text segment containing $m$ words is transformed into an $m * k$ vector matrix, which is equivalent to the image mapping of vectors.

$$
\begin{bmatrix}
x_{11} \cdots x_{1i} \cdots x_{1k} \\
x_{21} \cdots x_{3i} \cdots x_{3k} \\
\vdots \ \cdots \ \vdots \ \cdots \ \vdots \\
x_{m1} \cdots x_{mi} \cdots x_{mk}
\end{bmatrix}
\tag{11}
$$

The matrix $x_i$ denotes a word, and for a text of length $n$ can be expressed as:
$Text = x_{1:n} = x_1 \oplus x_2 \oplus x_3 \oplus \cdots \oplus x_n$, and the $i:i+m$ words are operated according to the formula to obtain the result of the convolution operation of the feature $C_i$.

$$
c_i = f(w \cdot x_{i:i+q-1} + b)
\tag{12}
$$

The results of $C_i$ are processed with a convolution kernel to obtain the features $c \in R^{n-q+1}$ of the corresponding layer. The most valuable features $\hat{c} = \{c\}$ are captured using the maximum pooling operation, and regularization is performed using dropout to finally obtain the text feature vector $Vectour_{Title}$ available for learning.

$$
Vectour_{Title} = droupt(\hat{c})
\tag{13}
$$

Multi-valued type attributes are handled by transforming the enumerable attribute values by means of index matrix and one-hot encoding. Suppose the item has a multi-valued attribute $Y$ and has $m$ enumerable attribute values, and the $m$ attribute values are

represented as a continuous number, then the attribute $Y$ can be represented as $[y_1, y_1, \cdots, y_m]$, and the embedding matrix is indexed by a sequence of *1-m*. For the *i*-th item $V_i$, the attribute values about its attribute $Y$ can be represented as a *d*-dimensional vector, and the $v_i$ attribute $Y$ of can be represented as

$$Vectour_{mutil}(v_i) = y_{i1} \oplus y_{i2} \oplus \cdots \oplus y_{in}. \tag{14}$$

The vectorization results of text types and multivalued attributes of the same element are merged, fed into the fully connected layer, and processed using the activation function to form an initial vector $v_{init}$ about the element.

$$v_{init} = f(w1(Vector_{Title} \oplus y_{i2} Vector_{mutil}) + b_1) \tag{15}$$

where $Vector_{Title}$ represents the vectorized result of the text type attribute of the item, and $Vector_{mutil}$ represents the vectorized result of the multi-value type attribute of the item.

### RS unit

The feature extraction unit performs the multimodal feature representation task and aggregates the vector results generated by multiple modalities as the initial vector of items in the subsequent recommendation unit to ensure that the important attribute value information is not lost. During the recommendation process, a vector matrix of users and projects is formed to provide input parameters for the cross-compression unit.

Assume that the attributes of a user can be represented as $U = [B_1, B_2, \cdots, B_x]$, where *Bi* represents an attribute domain of the user. The attribute values of the user's structured attributes are input to the embedding layer, and the user vector representation is obtained using Eq. (16) and the fully connected layer.

$$u = u_{B1} \oplus u_{B1} \oplus u_{B2} \oplus \cdots \oplus u_{Bx}. \tag{16}$$

### CCU unit

In the process of combining the knowledge graph with deep recommendation, CCU realizes the communication between the neighborhood graph embedding and semantic information embedding. The data from the two modules and the information contained in the entities they are associated with learn and improve each other. The recommended item data, and the triad data of the knowledge graph can be associated through this unit, and the triad vectorization and item vectorization results can be gradually improved through learning and training.

CCU is the linking module between $V$ and $E$. For the potential features $v_L \in R^d$ with $e_l \in R^d$, the cross-feature matrix $C_l$ representing the $L$-th level is constructed.

$$C_l = v_l e_l^T = \begin{bmatrix} v_l^1 e_l^d & \cdots & v_l^1 e_l^d \\ \vdots & \cdots & \vdots \\ v_l^d e_l^d & \cdots & v_l^d e_l^d \end{bmatrix}. \tag{17}$$

Eq. (18) describes the crossover operation in the crossover compression cell, form a matrix projection to obtain the final VE feature vector. The cross-compression cell can finally be represented as

$$[v_{l+1}, e_{l+1}] = C v_l, e_l. \tag{18}$$

### KG unit

In the D-KGR model, the triplet of the knowledge map is taken as the input parameter of the model, and valuable head and tail feature representations ($h_L$ and $t_L$) are extracted, respectively, through the processing links such as cross compression unit and multilayer perceptron, and the estimated $\hat{t}$ of the tail corresponding vector is obtained from Eq. (19) using the multi-layer neural network.

$$
\begin{aligned}
h_L &= E_{v \sim s(h)}\left[ C^L(v_{init}, h)[e] \right] \\
r_L &= M^L(r) \\
\hat{t} &= M^k \begin{bmatrix} h_L \\ r_L \end{bmatrix}
\end{aligned}
\tag{19}
$$

where rL: the initial feature of the relation, which is usually in a one-hot coding form. $S(h)$ denotes the set of header relationships. T is the item ID corresponding to the head vector $h$ in the data of the recommender system. $C^L$ represents the corresponding cross-matrix as introduced in the previous section.

Finally, the predicted score of the model can be calculated using Eq. (20).

$$f_{KG}(t, \hat{t}) = t^T \times \hat{t}. \tag{20}$$

In practical applications, we can use deep learning methods that integrate knowledge graphs and convolutional neural networks to improve the application effect of recommended learning resources. In this application, the convolutional neural network is divided into two parts: user and book, as shown in Fig. 7.

Generally speaking, data are divided into three types: single valued, multi valued, and text. For different data types, differentiated data preprocessing methods are adopted. The data processing approach for single valued attributes is to encode users or books, use convolutional neural network technology to form an embedding vector matrix, and reconstruct the knowledge graph to obtain the feature vectors of users and books. The user, book ID, gender, age, and occupation of the user are encoded using Onehot and are input into a user book fully connected layer. For the types of books, the combinations can be diverse; for example, the books can be both engineering and philosophy. Assuming there are a total of 18 types of books, a digital index can be used to connect the feature matrix and input it into the fully connected layer. The processing of book names is quite special and requires a text convolutional network. Compose the embedding vector of each word into a matrix, and then use convolutional kernels of different serial port sizes to perform
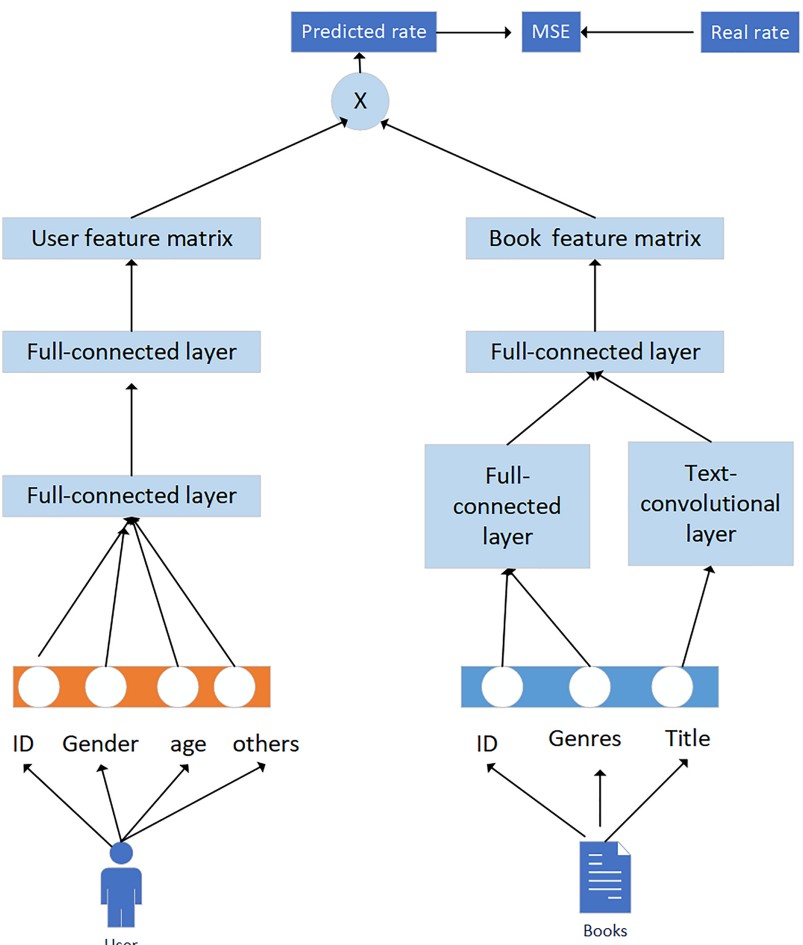

**Figure 7 Extracting user and book features using convolutional neural networks.**

convolution on the matrix, usually using $2 \times 2$, $3 \times 3$, $5 \times 5$. Then, through the pooling layer and using dropout for regularization, the book name embedding vector is obtained.

The specific processing flow is:

1. Users and books are processed using convolutional neural networks to extract user features. Two fully connected layers are used, with the first layer embedding different features into the same dimensionality $\times$ 128 vectors.

2. Different features (f) are connected and input into the second fully connected layer, so that the feature vector and book feature network maintain the same dimensionality $1 \times 200$.

3. For the book feature network, the name features of the book are extracted using a text convolutional neural network, converting words into word vectors, as shown in Fig. 8.

4. Convolutional filtering is applied and used to filter h word windows to obtain feature vectors. Next, the maximum pooling layer is used to extract important feature sentence information and connect the various feature vectors of the book. Then the fully connected layer is used to obtain a dimension of $1 \times 200$ book feature vectors.

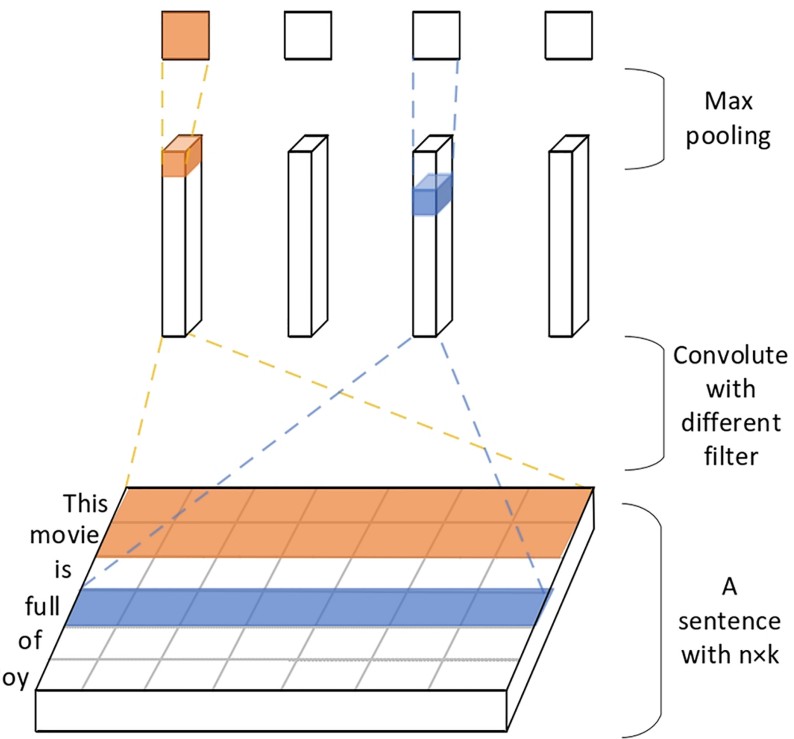

**Figure 8 Extracting book name feature vectors using text convolutional networks.**

5. The results of the user and book network mentioned above are multiplied and normalized with an actual rating; the loss function is optimized and the feature vectors of the user and book are obtained.

By reconstructing the knowledge graph, effective semantic information can be obtained. By utilizing cross compression units, their functions act like a bridge between the neighborhood graph embedding module and the semantic information embedding module. Information sharing and enhancement can be carried out between the project embedding representation and the corresponding head entity, which can mine higher-order relationships and semantic information in relationships. Moreover, more accurate recommendations can be made through alternating learning patterns of the modules.

The integration of deep learning and knowledge graphs can fully combine and utilize auxiliary information, enrich the description of users and items, enhance the mining ability of recommendation algorithms, and effectively compensate for the sparsity or lack of interactive information. The introduction of convolutional neural network algorithms into the knowledge graph helps resolve the cold start problem. The knowledge graph utilizes its expandable characteristics to carry the user's own attribute information when a new user enters the platform, forming a customized user background. The proposed system is able to determine the corresponding interestes of new users without any preference bias and also recommend similar products to new users.

# EXPERIMENTAL ANALYSIS

Although the improved BP algorithm based on particle swarm optimization has certain theoretical advantages, it needs to be verified experimentally. In this section, the simulation experiments are analyzed in terms of algorithm efficiency and accuracy.

## Algorithm efficiency analysis

The algorithm efficiency validation focuses on testing this algorithm against three personalized recommendation algorithms: particle swarm, neural network, and knowledge graph.

(1) Experimental environment and experimental data

Experimental environment: Bert-as-service (http://jmcauley.ucsd.edu/data/amazon/) launched by Tencent Technology, is an open-source AI laboratory, using a Win10 operating system, Python version 3.5 or above, TensorFlow software version 1.10, for the experimental simulation platform by pip command in the environment.

Experimental data: The Amazon Book dataset was used in this experiment (http://jmcauley.ucsd.edu/data/amazon/), which is currently mainly used for book recommendation. A total of 1,563 users and 2,503 projects were selected in the experiment (*Mcauley et al., 2015*), and interactions between users and projects were standardized at least six times to ensure the quality of the dataset.

(2) Comparative experiments

The data were used to conduct three experiments with each of the four recommendation algorithms (this algorithm, particle swarm, neural network, and knowledge-based algorithms). The averages were obtained which revealed variability in their time spent for different resource sets, and the comparison results are shown in Fig. 9.

(3) Experimental analysis

From the analysis of the data in Fig. 9, it can be concluded that with the gradual increase of the number of experiments and the amount of experimental data, the present algorithm has obvious advantages in terms of efficiency compared with the other three algorithms. When the number of resources involved in the test was only 1,000, this algorithm has no advantage in terms of running time. With the gradual increase in the number of tests, the advantages of integrating particle swarm iterative classification, neural network intelligent simulation and low resource consumption can be quickly processed for massive information data. Compared with other algorithms, the low complexity of the algorithm leads to low time overhead cost.

This algorithm is because in the process of processing knowledge graph relationships, rich semantic information is added to the aggregated neighborhood embedding, fully considering the structure of the graph. By mining higher-order relationships between neighborhoods through interaction, more effective interaction relationships between neighborhoods can be obtained, which can effectively construct user preference models without placing too much emphasis on deeper domain mining, thus avoiding the introduction of interference information. The algorithm has low complexity and achieves the goal of reducing resource consumption.

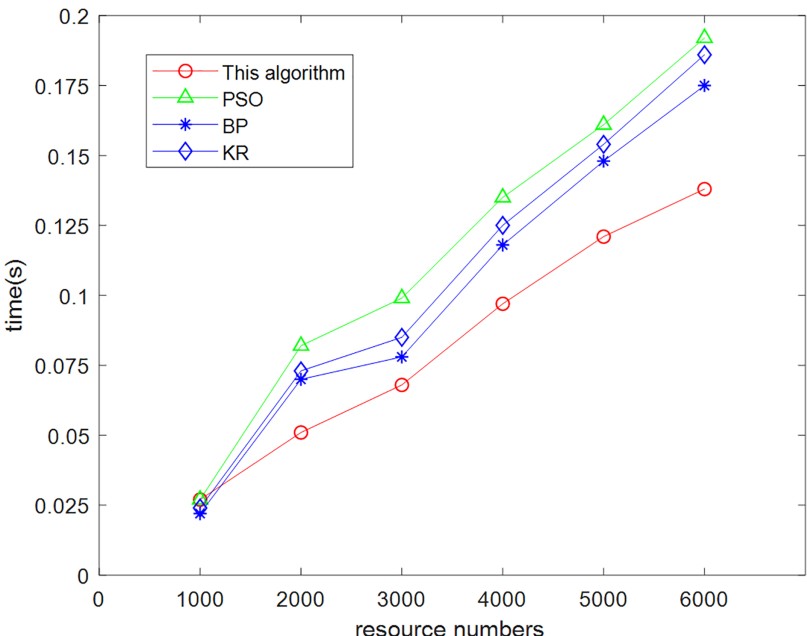

**Figure 9 Comparison chart of prediction time required by this algorithm and other algorithms.**

## Algorithm accuracy analysis

Additional experimental tests were needed to verify the accuracy of the algorithm's personalized recommendation results after the model was finalized. ZhihuRec (https://github.com/THUIR/ZhihuRec-Dataset) used a large-scale rich text query and recommendation dataset from Zhihu and Tsinghua University, with nearly 100 million exposures and the richest contextual information. The dataset is the largest interaction dataset for personalized recommendations. In this study, all data was divided into four samples with increasing numbers of target learners (40%, 60%, 80% and 100% of the total number, respectively); a total of 80% of the training set and 20% of the test set data was used, and the model was retrained using the training set data and validated using the test set data. The sample distribution is shown in Table 1.

There are various evaluation metrics used to measure the quality of recommendation algorithms, including accuracy metrics, metrics based on ranking weighting, coverage metrics, diversity and novelty metrics, and so on. Among them, the prediction accuracy metrics represented by root mean square error (RMSE) and mean absolute error (MAE) are widely used in recommendation algorithms. They are also frequently used to evaluate the accuracy of algorithms that are submitted in competitions. The root mean square error (RMSE) formula is as follows:

$$RMSE = \sqrt{\frac{1}{m}\sum_{i=1}^{m}(r - \hat{r}_i)^2}. \tag{21}$$

**Table 1 Sample distribution of target learners.**

|  | Sample 1 | Sample 2 | Sample 3 | Sample 4 |
| --- | --- | --- | --- | --- |
| Learners | 10,000 | 15,000 | 20,000 | 25,000 |
| Number of course resources | 800 | 1,200 | 1,600 | 2,000 |
| Number of ratings | 26,237 | 39,296 | 52,368 | 65,490 |

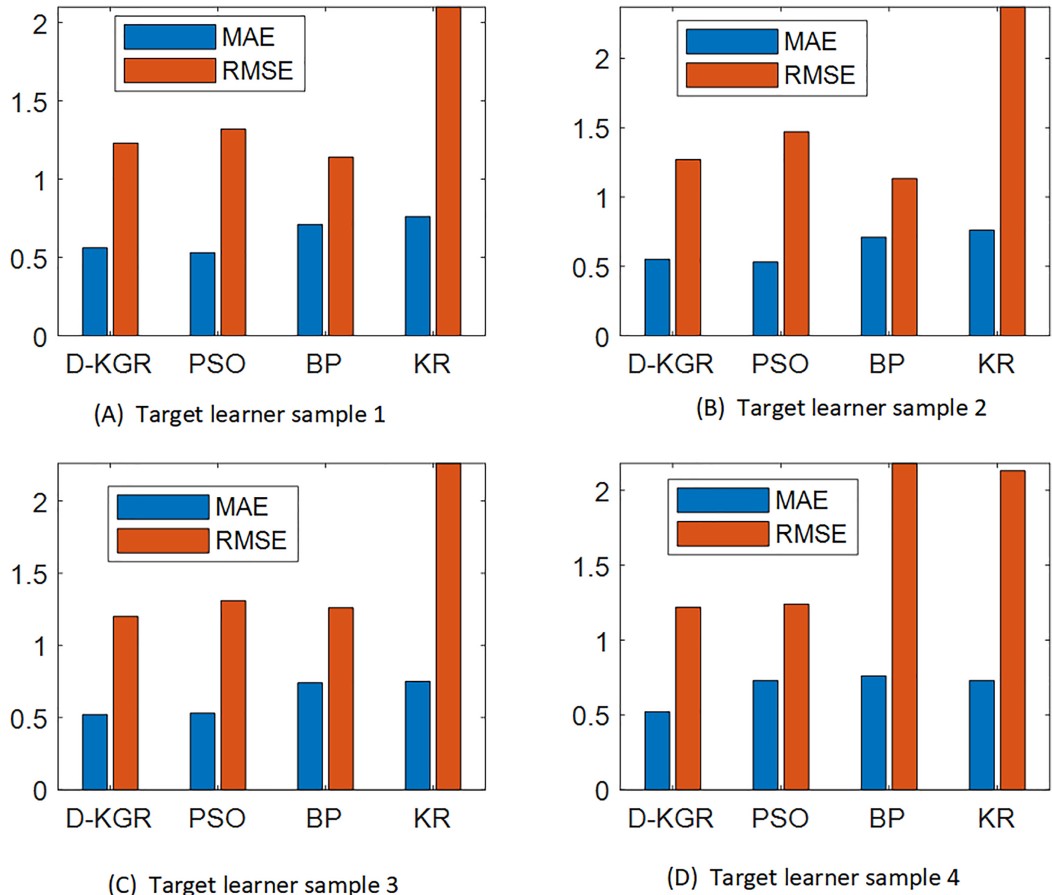

(A) Target learner sample 1

(B) Target learner sample 2

(C) Target learner sample 3

(D) Target learner sample 4

**Figure 10 (A–D) Comparative analysis results of MAE and RMSE values of four algorithms under different learning sample conditions.**

The mean absolute error (MAE) is as follows:

$$MAE = \frac{1}{m} \sum_{i=1}^{m} |r - \hat{r}_i|^2 \tag{22}$$

where $m$ is the test set size, $r_i$ is the user score, and $\hat{r}_i$ is the predicted score.

The particle swarm (PSO) (*Jiang, 2022*), neural network (NP) (*Qu et al., 2016*) and knowledge graph (KR) algorithms (*Li et al., 2022*) were selected for comparison in this experiment, and the results are shown in Fig. 10.

From the comparison results in Fig. 10, it can be seen that the D-KGR algorithm proposed here has lower MAE and RMSE values than the other classical algorithms for any group of four target learner samples participating in the experiment, A, B, C, and D. A large dataset has been found to reduce the accuracy between PSO, NP, and KR. These three models have alternating advantages in different ranges of K values, but overall, the KR model has a greater advantage, especially in sparse datasets (book datasets) where the advantages are more pronounced. In the case of limited user interaction data, the PSO model is restricted in the construction of historical interest sets for a user. Similarly, when the knowledge graph data is insufficient, the advantages of the NP model gradually decrease. The model in this study can combine the advantages of each of these methods, which is equivalent to using the knowledge graph for two mining operations simultaneously. This greatly compensates for the sparsity of the data and improves accuracy. The deep recommendation system algorithm based on knowledge graphs outlined in this study is very reasonable.

## CONCLUSIONS

Accurately recommending high-quality personalized learning resources is a major problem that learners are facing today due to the overwhelming amount of online material. This article proposes a new neural network deep learning algorithm based on the knowledge graph. Firstly, the knowledge graph matrix data was established. The knowledge graph and recommendation data were cross trained to integrate the feature representation of the knowledge graph in a multitasking manner. Secondly, we made full use of the historical interaction information between users and projects, introduced the convolution algorithm to conduct data cross processing, and then mined the implicit relationship between the user's feature representation and data through the user's context. Thus, we were able to provide target learners with a list of learning resources with the best matching degree. In the D-KGR model, the role of the knowledge graph was maximized. The head of the knowledge graph was alternately trained with project data, while the tail of the knowledge graph participated in the composition of the user's feature vectors. Joint training and alternating training were combined to improve the performance of the recommendation system. Future research should explore the knowledge graph and artificial intelligence combination, as well as the accuracy and efficiency of recommendations made under sparse data conditions.

### Funding

This work was supported by the National Natural Science Foundation of China (No. 61263033), the International Science and Technology Cooperation Project of Hainan (No. KJHZ2015-4) and the Higher School Scientific Research Project of Hainan Province (No. Hnky2015-80). The funders had no role in study design, data collection and analysis, decision to publish, or preparation of the manuscript.

## Grant Disclosures

The following grant information was disclosed by the authors:
National Natural Science Foundation of China: 61263033.
International Science and Technology Cooperation Project of Hainan: KJHZ2015-4.
Higher School Scientific Research Project of Hainan Province: Hnky2015-80.

## Competing Interests

Xiao Feng Ma is an employee of Chengfang Technology Co., Ltd.

## Author Contributions

- Yan Wang analyzed the data, prepared figures and/or tables, authored or reviewed drafts of the article, and approved the final draft.
- Xiao Feng Ma performed the experiments, performed the computation work, prepared figures and/or tables, authored or reviewed drafts of the article, and approved the final draft.
- Miao Zhu conceived and designed the experiments, authored or reviewed drafts of the article, and approved the final draft.

## Data Availability

The computer code is available in the Supplemental File and at https://cseweb.ucsd.edu/~jmcauley/datasets.html#amazon_reviews and https://github.com/THUIR/ZhihuRec-Dataset.

## Supplemental Information

Supplemental information for this article can be found online at http://dx.doi.org/10.7717/peerj-cs.2010#supplemental-information.

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
