# Peer review of "A knowledge graph algorithm enabled deep recommendation system"

_PeerJ Computer Science, doi:10.7717/peerj-cs.2010_

## Round 0.1 · original submission · Major Revisions

As per comments from reviewers, I suggest a major revision for this paper.

**Language Note:** The review process has identified that the English language must be improved. PeerJ can provide language editing services - please contact us at [email protected] for pricing (be sure to provide your manuscript number and title). Alternatively, you should make your own arrangements to improve the language quality and provide details in your response letter. – PeerJ Staff

Reviewer 1 ·

Basic reporting

check comments

Experimental design

check comments

Validity of the findings

check comments

Additional comments

• What motivated your research in the field of personalized learning resource recommendation?
• Could you explain the challenges associated with existing personalized recommendation systems in the context of online learning?
• How does your proposed algorithm leverage knowledge graphs to enhance personalized recommendations?
• What benefits does the integration of knowledge graphs bring to the recommendation process?
• Can you elaborate on how deep learning algorithms are used to extract data features for recommendation?
• What specific deep learning techniques or models are employed in your algorithm?
• How do convolutional methods contribute to the data processing in your algorithm?
• Could you provide examples of how these methods improve the pertinence of recommended learning resources?
• You mentioned that your algorithm has low complexity and resource consumption. Could you explain how this efficiency is achieved?
• What are the advantages of low time and sales costs in the context of personalized learning resource recommendation?
• Can you share some insights into the performance of your algorithm when learning resources and users reach a certain scale?
• Author can read the following papers to increase the technical strength of the paper:Scholar recommendation based on high-order propagation of knowledge graphs, Recommendation of Healthcare Services Based on an Embedded User Profile Model, A personalized approach to course recommendation in higher education, Automated and Personalized Nutrition Health Assessment, Recommendation, and Progress Evaluation using Fuzzy Reasoning

• What metrics or criteria were used to evaluate the accuracy and effectiveness of your recommendation system?

Cite this review as

Reviewer 2 ·

Basic reporting

Authors should improve overall English standard of this manuscript.

Experimental design

Contributions and novelty level is low.

Validity of the findings

Authors are highly recommended to provide references of PSO, BP and KR in section 4.

Additional comments

Main queries are as follow:

Major contributions are limited, authors should work more on proposed methodology.
Comparative analysis is weak and lack any significant explanation. Authors should modify section 4.
Contribution and abstract needed to be concise and to the point.

Cite this review as

Reviewer 3 ·

Basic reporting

The meaning of the research title is too broad. The author should improve the title.

In the abstract, the result of this work must be described briefly with data. The result of this work is not clear.

I miss comprehensive literature about “existing learning resource recommendation algorithms”.

The quality of the figures should be improved.

Please rewrite all the equations. Besides, unify the font size of all equations.

The authors claim that “It can quickly process massive information data, and the low complexity of the algorithm leads to low time and sales costs. It has obvious advantages in algorithm efficiency”. I missed a short study about time complexity regarding the proposed algorithm.

Experimental design

The effectiveness of this work is not clear. Through experiments, the authors must justify the effectiveness of the proposed method by comparing it with state-of-the-art related methods. No comparison is shown with related state-of-the-art learning resource recommendation methods from recent literature. Comparative analysis needs to be included in the experiments and discussion section.

Validity of the findings

Furthermore, it would be advantageous and interesting to demonstrate the model's reliability by including accuracy metrics such as nDCG, Recall, or Precision. No less important would be to analyze how the model behaves in terms of diversity and novelty concerning the items in the catalog. Last but not least, it would be valuable to investigate bias metrics (REO and RSP).

Future work needs to be included, which helps the readers further extend the proposed method.

Cite this review as

---

## Round 0.2 · Minor Revisions

I suggest a minor revision for this revised paper as per comments from original reviewers.

Reviewer 2 ·

Basic reporting

Satisfied

Experimental design

Satisfied

Validity of the findings

Satisfied

Additional comments

Satisfied with revised version.

Cite this review as

Reviewer 3 ·

Basic reporting

Thank you for your continued work on this paper. I appreciate the effort you have put into it. However, I have noticed that some of the comments provided have not been fully addressed in the latest revision. Specifically, in the Introduction section, it would be valuable to elaborate on the challenges associated with existing personalized recommendation systems in the context of online learning, providing examples for reference. Addressing this point will significantly enhance the comprehensiveness of the manuscript.


To further elevate the overall quality of the paper, I would like to underscore the importance of refining the clarity and precision of your English. There are a few instances where language improvements could enhance overall comprehension. Your attention to these details will undoubtedly contribute to the overall strength and impact of your work.

Experimental design

I would like to suggest some additional considerations that could enhance the overall quality and impact of your research. It would be advantageous and interesting to bolster the demonstration of your model's reliability by incorporating accuracy metrics such as nDCG, Recall, or Precision. This would provide a more comprehensive evaluation and offer valuable insights into the model's performance.

Validity of the findings

I would like to suggest some additional considerations that could enhance the overall quality and impact of your research. It would be advantageous and interesting to bolster the demonstration of your model's reliability by incorporating accuracy metrics such as nDCG, Recall, or Precision. This would provide a more comprehensive evaluation and offer valuable insights into the model's performance.

Cite this review as

---

## Round 0.3 · accepted · Accept

In the opinions of two original reviewers, this revised paper can be accepted for publication now.

Reviewer 2 ·

Basic reporting

Satisfactory.

Experimental design

Satisfactory.

Validity of the findings

Satisfactory.

Additional comments

None

Cite this review as

Reviewer 3 ·

Basic reporting

no comment.

Experimental design

no comment.

Validity of the findings

no comment.

Cite this review as